# Effect of out-of-school visual art activities on academic performance. The mediating role of socioeconomic status

Genman Deer[1], Hao Wu[2‡], Li Zhang[3], Endale Tadesse[4], Sabika Khalid[4], Congyu Duan[5], Wang Tian[6], Chunhai Gao[7] *

1 Faculty of Education and human development, The Education University of Hong Kong Faculty of Education and Human Development New Territories, Taipo, New Territories, Hong Kong, 2 School of Education Science, Nanjing Normal University, Nanjing, China, 3 Faculty of Education, Liaoning Normal University, Dalian, China, 4 College of Teacher Education, Zhejiang Normal University, Jinhua, China, 5 The Greater Bay Area Institute of Educational Research, Shenzhen University, Shenzhen, China, 6 Professional Higher Education college, Yunnan Arts University, Kunming, Yunnan Province, China, 7 Faculty of Education, Shenzhen University, Shenzhen, China

‡ HW is the co-first author of this article.
* chunhaigao@hotmail.com

**Data Availability Statement:** All relevant data are within the manuscript and its Supporting Information files.

## Abstract

The application of visual art and other extracurricular activities to children's sustainable development is predominantly discussed in Western countries. Consequently, non-Western society could not cherish the benefit of visual art on their children's cognitive and non-cognitive skill development due to a lack of evidence that would revive the community, educators, and policy-makers' impressions about visual art activities, in addition to its amusement use. Thus, the present study adopted a cross-sectional study comprised of a large-scale survey (N = 1624) taken from the southwest part of China to assess the impact of out-of-school visual art activities on children's academic attainment across economically advantaged and disadvantaged children. Astonishingly, the study's findings shed light on current Chinese parents' dedication to purchasing out-of-school activities regardless of their social class difference; notwithstanding, lower-class parents ought to learn that spending time with their children during their activities is more beneficial. The study's implication calls for curriculum policy reform involving aesthetic education and expanding community youth centers for different extracurricular activities.

## Introduction

There is a rich body of evidence about the application of in-and-out-of-school extracurricular activities (ECAs) in children and adolescents' cognitive and noncognitive adjustments [e.g., 1–4]. Western countries provide ECAs (sport, music, theater, scouts) after formal classes, which are supervised by an adult so that students can enhance their physical, social, psychological, and academic well-being [3, 5–8]. In-school ECAs are mostly known as "after-school programs" since children are involved in various ECAs with the same peers they meet in the

**Funding:** The authors received no specific funding for this work.

**Competing interests:** The authors have declared that no competing interests exist.

classroom, although out-of-school ECA programs include children with another group of peers with different skills, communication, and behaviors [5, 8, 9]. Marsh and Kleitman classified out-of-school ECAs as structured activities under formal training, adult supervision, and leisure activities primarily for fun that do not have an intended outcome [10]. In this paper, out-of-school (OOS) activities represent structured and organized activities that require sincere interest and effort to develop oneself [2, 11]. In addition, most developing countries with limited educational resources cannot promote the breadth of ECAs inside the school, so parents must enroll their children in privately-owned schools that service several ECAs based on the student's and parent's choice [1, 12]. Such ECAs, well organized, resourced, and intended, are more likely to boost children's academic and non-academic outcomes [4, 13].

Furthermore, children who participate in OOS ECAs outdo their counterparts in social adjustment, psychological well-being, and academic performance [5, 14], whereas low-income children spend their after-school time with activities that have no substantial benefit for them, perhaps destroying their future [9, 15] such as watching TV, playing games [8]. Although many studies have highlighted the robust impact of OOS ECA participation on children's development, it is still controversial to generalize its generic effect due to the paucity of concrete evidence [e.g., 7, 16, 17]. Regardless of children's SES, the likelihood of participating in several ECAs depends on their gender; boys are primarily interested in ECAs such as sports, dancing, and band/music, and girls prefer visual arts, theater, and music instruments [18, 19]. Boys stereotype painting, drawing, and music training as girlish, and girls feel sports activities, dancing, and playing an instrument are masculine [3].

Furthermore, an extensive body of literature has discussed ECAs as a whole, which has vague implications for policy-makers who do not specify the corresponding advantage of ECAs socially, academically, psychologically, or physically [e.g., 6, 8, 20]. For instance, in- and OOS sports activities are related to never using substances [10, 21]. Likewise, it is well documented that children's high number of ECA participation leads to unintended or minor outcomes [c.f., 14, 22]. A later study in China stipulated that time spent attending ECA tutoring on weekdays was negatively associated with students' cognitive development [9]. Most importantly, the present study ought to fill the gap that previous studies considered the purpose of ECAs by only assessing in-school activities, ignoring OOS activities. Hence, in view of children's development outcomes, this paper sought to examine the participation in OOS visual art activity, which is considered a pertinent ECA for children's academic performance [2, 23]. In addition, previous evidence measures organized ECA participation only by considering formal activities. Still, it should also be assessed through childhood and adolescent involvement with their parents, in addition to organized OOS formal participation. Emerging literature in the contemporary world has argued that the effect of ECAs is more pronounced when parents devote their time in and out-of-home ECAs in addition to the formal ECA program [1, 24].

Consequently, children's socioeconomic status (SES) determines their participation in formal and informal OOS ECAs [18, 25]. Nevertheless, emerging literature has claimed that, except for family SES, participating in art lessons does not significantly affect a child's development [12, 25]. Hence, parents need a decent income to afford formal adult-supervised OOS ECA training and dedicate their time to activities that strengthen children's ECA skills and develop parent-child attachments [1, 26]. A follow-up analysis by Lagacé-Séguin and Case indicated that parents who support their children alongside the paid OOS program foster their academic outcome [22]. Likewise, in addition to paid OOS training, children develop robust cognition and emotion for visual art, and more parents commit and are involved in artistic work with their children [9, 12, 27]. Therefore, parents should dedicate their time and money to obtain the intended outcome of visual art and other ECAs [2].

Stimulatingly, although SES determines children's participation in OOS ECAs, a growing number of studies claim that low-SES or disadvantaged children who are minority participants in in-and-out-of-school organized and supervised activities benefit more than their high-SES peers [1, 27, 28]. Explicitly, a longitudinal study in the USA that assessed the significance of ECAs stated that from all activities, visual art affects the well-being of low-SES counterparts compared with high-SES counterparts [29]. Nevertheless, some particular evidence explains the effect of participation in OOS visual art activities on disadvantaged or low-SES children's academic adjustment in non-Western countries where this population is more prominent. Thus, to contribute concrete national and international literature and provide robust implications for stallholders, the current study operates on visual art participation, measured as supervised and organized OOS and informal visual art activities in and out of home and supported by parents.

## Literature review

### Visual art activities and disadvantaged children

Visual art amusement activities, and most children or young adolescents engage in non-intended outcomes or benefits [20, 30]. Nonetheless, in the contemporary era, visual art activities have received considerable attention among scholars, educators, and policy-makers in Western countries [20, 31]. Visual art represents a series of activities that include painting, sculpture, designing, drawing, photography, etc. [20, 30, 32]. For young adolescents, rather than physical and instrumental activities, visual art activities are the most engaging and creative activity that makes children widen their imagination and outlook on their surroundings [33, 34]. Similarly, visual art is a psychological intervention for children's traumatic experiences to reduce anxiety and restore self-efficacy [12, 35]. Accordingly, visual art education has a substantial advantage for children to develop social communication with peers, self-efficacy in their potential or cognition, and motivation for their school work [12, 23, 36]. Astonishingly, from different ECAs servicing in-and-OOS academics, high-flier students prefer to participate in visual art activities [18] given that these students possess higher SES or cultural capital [11, 37]. Shockingly, you can count the number of studies conducted to examine the impact of participating in OOS visual art activities on in-school academic achievement.

Conway's literature review demonstrates that children possess an accessible learning environment with no right or wrong during visual art activities. This learning freedom empowers children who are learning a language [23]. Thus, students, parents, and teachers consider visual art activities as a means of children's cognitive development and school learning outcomes [12, 36]. Remarkably, academically challenged, disadvantaged, and low-income children have an extraordinary chance of being affected by visual art activities compared to affluent children [20, 32, 34, 38]. Mansour and colleagues affirm that the worth of art is limited to fostering marginalized children's academic achievement and building a healthy personality [18, 38]. Concurrently, recent evidence stated that low-SES parents send their children to OOS visual art programs not only so their children can develop solid artistic skills [32, 37] but also for vigorous reasons that lay a strong foundation for their future life [25].

On the other hand, it is known that high-SES children have adequate exposure to different ECA equipment at home, which weakens the impact power of OOS ECAs for high-SES children [11, 33]. In light of visual art's undisputable advantage, it is a feasible intervention for at-risk children and adolescents to enhance their school learning difficulty and psychological issues [35, 38]. As a result, visual art has a promising future in the educational system to close the academic achievement gap due to children's out-of-school factors. A recent survey that used music and visual art intervention on improvised children (Hispanic) showed that parents

reported radical changes in their children's art interests, personal behavior, and academic achievement after eight months of training [32].

## China and out-of-school ECAs

China requires ECAs to endorse children and adolescent well-being, although the Confucius philosophy teaching and parenting style has made children reserved for expressing emotion, which is significantly costing the country [24, 39, 40]. The nation is calling for a society that cherishes the application of ECAs for sustainable children and adolescent development [1, 41]. Similarly, Chinese parents send their children to OOS programs for various benefits depending on their age and school level. For instance, parents with primary and high school children prefer ECAs that promote academic achievement [16, 42]. Nevertheless, scholars have been reminded of the scarcity of literature regarding the participation of contemporary Chinese children in OOS ECAs after school [1, 42]. Chinese literature discusses extracurricular activities as whole or nonvisual art activities (sports and music), which have rare effects on children's academic development [1, 2, 7, 43, 44]. A recent study on kindergarten children demonstrated that the more ECAs children attend, the more they lose their affection for it and the less they benefit from it [39]. Likewise, Hui and colleagues' experimental study affirms that children who attend visual art solely outperform their counterparts who attend training that integrates art with other ECAs [43]. Likewise, a later study on preschoolers' participation in several ECAs elucidates that the number of ECA children participating is unrelated to any expected outcomes [16, 45]. Above all, the current study explicitly aimed to study the association of children's academic performance regarding their participation in structured OOS visual art activities at privately owned centers and visual art activities supervised and involved by parents. Apart from that, this paper aimed to examine the relationship between participation in OOS visual art activities and parental socioeconomic status and whether the benefit of OOS visual art participation is more significant for disadvantaged children than for affluent children. Considering the possible advantages of visual art for underprivileged children, it is crucial to identify disadvantaged children in Chinese society.

## The power of socioeconomic status in contemporary China

In China, student academic achievement is the primary indicator of one's future life path [2, 46, 47], so Chinese parents are known for valuing children's school performance [16, 39, 40, 48]. Due to the meritocratic political ideology of China, parents primarily want their children to spend their leisure or after-school time attending privately owned tuition centers or studying at home, which is based on cultural capital [1, 24, 48]. However, recently, the government banned every privately owned tuition center from giving shadow education, which is a creative social injustice and children's academic pressure [9]. The Chinese government promotes commercial schools that provide extracurricular activities to foster academic and non-academic well-being [6, 45]. Notwithstanding, unlike Western countries, in China, OOS ECAs are serviced by privately-owned training centers that require a hefty fee [1, 24, 45], which still reflects the direct influence of SES on access to social assets and its outcomes [c.f., 40, 47, 48]. Unfortunately, Chinese parents are intensively busy with their jobs and cannot involve their children in OOS ECA participation in addition to paying the fees [16, 46]. Western countries' literature has emphasized the substantial influence of parental time deviation in in-and-out-of-home ECAs and the magnitude of advantages children can attain from commercial OOS training. Surprisingly, a growing body of evidence in China indicates that middle- and high-income parents currently show rigorous support, involvement, and expectations for their children's academic and non-academic engagement [e.g., 39, 40].

What is the power of SES on children's educational attainment in contemporary China? Harmoniously, the latest meta-analysis claimed a strong correlation between SES and Chinese students' language performance [47]. Subsequently, it is underlined that the impact of SES on children's academic performance is more indirectly prominent [40, 48] through OOS ECA [7]. A structural equation model suggested that it is essential to study potential factors that outshine the weight of SES on children's cognitive development [2]. Ren et al. administered a longitudinal study on preschoolers in Shanghai, which stated that the involvement of children in ECAs boosts their cognitive development and language learning skills [6]. Although emerging evidence has explored the application of OOS ECAs, a plurality of them are condensed in prosperous cities such as Hong Kong, Beijing, and Shanghai [6, 24, 45], where society does not have the financial constraints to send their children to OOS ECA centers. As a result, the robust effect of SES on the likelihood of participation in OOS activities is unfolded [19]. In addition to the affluent cultural capital of these cities, society is fully aware of and has witnessed the advantages of the ECAs their children obtained [11, 24]. However, the evidence pronounced that affluent parents send their children to OOS commercial centers for several ECAs for trivial purposes [42]. Thus, it is reasonably necessary to investigate children's participation in OOS ECA in other provinces or cities in China where there is a wide SES disparity [12, 47] and assess the threat and opportunities of OOS ECA for disadvantaged society. Flores stated that disadvantaged or low-income children with severe financial struggles attend OOS programs since parents know the superlative benefit children can obtain through organized activities that can be accessed at home, unlike high-SES peers [32]. Moreover, previous evidence in China is concentrated on early childhood participation in school ECAs, making our study the only one to address such a relevant concept at the primary school level for national and international academic society. Hence, to the best of our knowledge, this study is the first paper in China to investigate the impact of participation in OOS visual art activities (organized and non-organized) on disadvantaged children's academic performance. The paramount aim of this study is to examine the impact of formal and informal visual art activities on children's academic performance and the mediating role of parental SES to assess the role of SES on contemporary Chinese in predicting the likelihood of participation in visual art activities and academic performance.

## The present study

After a comprehensive literature review, the current study proposes the following hypotheses: (a) OOS visual art activities directly influence children's academic performance. (b) The impact of parents' participation in OOS visual art activities on children's cognitive development is more extensive than in OOS visual art activities organized by commercial training centers (c) The socioeconomic status of children is related to the likelihood of participating in OOS visual art activities (d) Economically disadvantaged children who participate in OOS visual art activities benefit more than their high-SES counterparts in terms of cognitive development. These questions enable us to assess the possibility of OOS visual art activities closing the academic achievement gap due to social class. Thus, unlike most national and international literature on early childhood samples, which is unfeasible for assessing the association of OOSECAs and cognitive development, this paper administered an extensive survey on public primary school children. In addition, the present study noticed that a vast body of Chinese literature concerning OOS ECAs centered in a few wealthy cities where the power of SES cannot be generalized to other cities and provinces; therefore, this paper sought to target Guizhou located in the southwestern part of China, where the SES distribution is spread out.

## Theoretical framework

Based on the primary aim of the present study, we adopted *the development model* and *social inequality gap reduction* model [10]. The *development model* demonstrates that ECAs' participation significantly contributes to children's social, psychological, and cognitive development [6, 10, 20, 49]. A wealth of international evidence indicates that ECAs, including visual art, promote a student's academic performance [e.g., 10, 13, 45]. Explicitly, in Mansour's longitudinal model, visual art notably impacts children's academic performance after adjusting for socio-demographic factors and prior academic achievement [20]. Seow and Pans' review exemplifies three ways ECAs indirectly foster children's cognitive development [49]. First, children develop self-esteem, self-concepts, and artistic discipline, which benefit academic learning, self-control, and persistence. Second, children participating in ECAs develop solid communication skills with their peers and an attachment to their school environment [12]. Third, participation in ECAs opens the door for children to create a means to academically join oriented peer groups that leads them to perform better than others through the social capital they obtain [6].

The second model proposed to guide the study is *social inequality gap reduction*, suggesting that ECA involvement is more beneficial to socioeconomically disadvantaged children than advantaged children [10, 20]. This model has a promising perspective in closing the academic gap between economically disadvantaged and advantaged students through ECAs [10, 20]. A stimulating volume of studies found that academically, economically, and psychologically marginalized or disadvantaged children acquired considerable benefits from ECAs [10, 28, 50]. The literature mainly elucidates that from other ECAs, visual art activities are latent to supporting disadvantaged children in improving their academic performance [35, 38]. Hence, this paper examined the extent to which OOS visual art activities sound the weight of SES on accessing social capital and the possibility that SES closes the academic achievement gap between economically disadvantaged and advantaged children.

## Method

### Participants

The present study adopted a cross-sectional research design that randomly sampled 1624 primary school students (52.5% girls) from 32 public schools in Guizhou Province in southwestern China. Power analysis for sample size determination is performed, and the effect size was moderate effect (F = 0.25), the significance level was $\alpha = 0.05$, and the power was 0.80. A primary school in Chinese educational structure represents children aged 7 to 14 attending grade levels from 1 to 6. In our sample, the average age of the primary children was (M = 11.6, SD = 2.03) (See Table 1). Informed written consent was obtained from each participant's parents/ guardians before participating in the study. Later, with the help of school teachers and administrations, we insisted that students attempt the questionnaire, which intends to measure several inquiries. The Ethics Committee of Liaoning Normal University approved this study (LL2022040). Initially, before dispatching our questionnaire, the parents of 2,500 primary school children selected from all primary schools in the province were asked to return informed consent forms. After waiting two weeks, we obtained consent from 2,106 parents for their children to participate in the study. Then, we dispatched a survey written in Mandarin for each grade level of sampled children with substantial support from teachers and head teachers. After collecting the paper-based questionnaires, we omitted 392 unfinished questionnaires with missing or inappropriate responses, yielding a final dataset containing 1624 responses. All the 1624 sampled participants were required to consult their parents in advance about their household socioeconomic status, including educational level and monthly income.

**Table 1. Demographic information of the sample.**

| Variables | M(SD) % |
|---|---|
| *Children Age* | 12.6 (2.03) |
| *Gender (Female)* | 852 (52.5%) |
| *Percentage of an only child* | 14.9% |
| *Percentage of at least one parent migrant* | 53.3% |
| *Marital status* | |
| Married | 80.1% |
| Divorce | 10.5% |
| Remarried | 7.2% |
| Widowed | 2.2% |
| *Mother educational level* | |
| Did not finish primary school | 13.1% |
| Primary School | 21.4% |
| Junior high school | 36.9% |
| Vocational High school | 6.5% |
| Senior high school | 15.2% |
| Undergraduate and above | 6.8% |
| *Father Educational Level* | |
| Did not finish primary school | 8.6% |
| Primary School | 19.4% |
| Junior high school | 40.0% |
| Vocational High school | 7.4% |
| Senior high school | 16.4% |
| Undergraduate and above | 8.2% |
| *Mother monthly income* | |
| Under 1000RMB | 28.8% |
| 1000-3000RMB | 44.0% |
| 3000-6000RMB | 20.6% |
| 6000-9000RMB | 4.4% |
| Over 10000RMB | 2.2% |
| *Father Monthly Income* | |
| Under 1000RMB | 16.1% |
| 1000-3000RMB | 40.4% |
| 3000-6000RMB | 28.5% |
| 6000-9000RMB | 11.0% |
| Over 10000RMB | 4.0% |

## Measurements

**SES.** The socioeconomic status of our participants was measured with two common indicators: parents' educational levels and monthly income. A large number of studies measure parental educational levels (father and mother) with 4- to 5-point Likert scales. However, the present study discusses the low SES province where society exhibits a diverse range of parental educational attainment. Accordingly, parental educational level (for both fathers and mothers) was captured with the following 7-point Likert scale: 1 = did not finish elementary school (8.6% of fathers and 13.1% of mothers), 2 = completed primary school (21.4% and 19.4%, respectively), 3 = completed junior high school (36.9% and 40.0%, respectively), 4 = completed vocational high school (6.5% and 7.4%, respectively), 5 = completed regular senior high school

(15.8% and 14.5%, respectively), 6 = completed junior college (15.2% and 17.4%, respectively) and 7 = completed a university undergraduate or postgraduate degree (6.8% and 8.2%, respectively).

The distribution of parental educational levels confirms our speculation that rural Chinese parents' educational levels are not normally distributed. The other SES indicator is monthly parental income, which was recorded from responses by each participating student's parents or legal guardians (grandparents) on a five-point Likert scale: 1 = under 1000 RMB, 2 = 1000–3000 RMB, 3 = 3000–6000 RMB, 4 = 6000–9000 RMB, and 5 = over 9,000 RMB. The internal consistency reliability coefficients (Cronbach's alphas) for the parental income of the fathers and mothers of our participants were 0.67 and 0.65, respectively.

**Visual art participation (VAP).** The magnitude of children's OOS visual art participation is measured from two aspects: (VAP1) participation in OOS visual art activities organized and supervised by privately owned training centers, and (VAP2) participation of OOS visual art activities supported and involved by parents in-and-out of home activities (C.f., 1). We designed 12 items that assess children's involvement in OOS art schools and visual arts activities for both dimensions. The items were rated on a 5-point Likert scale ranging from 1 = = strongly disagree to 5 = = strongly disagree. Sample items were as follows: "I often go to art exhibitions or community cultural activities," "I like all kinds of courses related to art," and "I draw many visual elements such as lines, shapes, spaces, and colors".

**Academic performance.** Given that the participants in the present study were elementary school students in grades 1 to 6, we could not use scores from standardized tests. Therefore, given that the final examinations in different schools and grades are different, we standardized all the test scores to eliminate the potential impact of school and grade on the academic scores. We standardized the scores for the Chinese, English, and math subjects using the score distribution in the specific school and the student's grades. All subsequent analyses utilized these standardized scores as the children's test scores.

## Research findings

### Descriptive analysis

A two-way multiple analysis of variance (MANOVA) was administered to examine the mean difference between the participants' SES and gender in the OOS visual art activity and academic performance. Thus, the analysis examined the possible association between the measured SES, gender, OOS visual art participation, and academic performance. According to the multivariate analysis results presented in Table 2, there was no significant participation difference in VAP1 and VAP2 among boys and girls ($F$ (1) = 2.246, $P$ = .071, and $F$ (1) = 3.77, $P$ = .31, respectively). In addition, the Table 2 results indicate a surprising result that there is no notable academic performance difference among boys and girls in Chinese ($F$ (1) = 2.506, $P$ = .061), English ($F$ (1) = 1.573, $P$ = .31), or math ($F$ (1) = 3.550, $P$ = .094). Astonishingly, except for the mother's income ($F$ (4) = 12.78, $P$< .05, η2 = 0.06), although the MANOVA result demonstrates that none of the parental levels of education or father's monthly income determines the likelihood of participation in VAP1, the father's educational level ($F$ (4) = 12.78, $P$< .05, *η2* = 0.18) and mother's monthly income ($F$ (4) = 12.98, $P$< .01, η2 = 0.21) are associated with children participating in VAP2. For further understanding, we ran the Turkey Post Hoc test to explain the mean difference under each variable category. The *Post Hoc* test noted that the mean VAP2 score difference between fathers with no primary education and undergraduate holders was MD = -0.66 (SE = 0.129). Moreover, the test result stated that the mean difference of VAP2 between mothers with an income under 1000 RMB and 6000–99000 RMB was (MD = -0.3719, SE = 0.1035). Similarly, unlike the father's SES, the mother's income and

**Table 2. Two-way MANOVA result.**

| | VAT1 | | VAT2 | | Chinese | | English | | Math | |
|---|---|---|---|---|---|---|---|---|---|---|
| | F | SE | F | SE | F | SE | F | SE | F | SE |
| Sex | 2.246 | .037 | 3.77 | .042 | 2.506 | .603 | 1.573 | .857 | 3.550 | .788 |
| FE | 1.728 | .053 | 12.78* | .068 | 1.446 | 1.853 | 1.361 | 1.902 | 1.912 | 1.595 |
| ME | 1.671 | .760 | 1.268 | .122 | 12.1** | .340 | 21.47** | .456 | 11.97** | .397 |
| FI | 1.403 | .066 | 2.147 | 1.340 | 1.243 | 1.767 | 2.314 | 2.508 | .615 | 1.095 |
| MI | 12.78* | .007 | 12.98** | .050 | 10.37*** | .420 | 15.55*** | .570 | 17.56** | .519 |

Note

*$p < .05$

***$p < .001$. ME = Mother Education level; FE: Father Education level; FI = Father Monthly income, MI = Mother Monthly Income; Gender code = 1: Male, 2: Female

educational attainment were related to the academic performance of children in Chinese (($F$ (6) = 10.37, $P < .001$, η2 = 0.28, $F$ (6) = 12.1, $P < .05$, η2 = 0.15, respectively), English (($F$ (6) = 15.55, $P < .001$, η2 = 0.31, $F$ (6) = 21.47, $P < .01$, η2 = 0.08, respectively), and Math (($F$ (6) = 17.56, $P < .01$, $F$ (6) = 11.97, $P < .01$, η2 = 0.19, respectively).

Subsequently, as presented in Table 3, a bivariate analysis was administered to assess the noncausal relationship among the operating variables. According to Pearson's correlation findings, participation in VAP2 had a robust association with children's academic performance ($r = .731$, $P < .001$), SES ($r = .690$, $P < .001$) and the likelihood of participating in VAP1 ($r = .783$, $P < .001$). Surprisingly, the bivariate result indicates that primary school students' age range has no significant influence on their participation in VAP1 ($r = .204$, $P = .082$) and VAP2 ($r = .301$, $P = .304$).

## The effect of participation in OOS visual art on economically disadvantaged children

To meet the primary aim of the present study, we adopted a path analysis or a structural equation model to assess the children's OOS visual art activity participation (VAP1 and VAP2) direct and indirect effects on their academic performance with the mediation of SES. Hence, we performed a structural equation model (SEM) with IBM AMOS 21.0 on a dataset with no missing values. Structural equation modeling (SEM) or path analysis was performed using AMOS 21.0 to examine the mediating role of the speculated variables. According to the output of the SEM, the results indicate a good model fit. The chi-squared $\chi^2$ (4) = 1542.87, goodness-of-fit index (GFI, 0.765), comparative fit index (CFI, 0.178), and the mean square error of

**Table 3. Bivariate correlation among variables.**

| | M | SD | α | 1 | 2 | 3 | 4 | 5 |
|---|---|---|---|---|---|---|---|---|
| Age | 12.6 | .952 | .781 | | | | | |
| AP | 77.915 | 13.19 | .701 | .261 | | | | |
| SES | 2.7474 | .891 | .824 | .548* | .671*** | | | |
| VAP1 | 3.0370 | .767 | .851 | .204 | .521*** | .492*** | | |
| VAP2 | 3.1497 | .874 | .843 | .301 | .731*** | .690*** | .783*** | |

Note

*p < .05

***p < .001, AP = overall Academic performance, SES: computed socioeconomic status

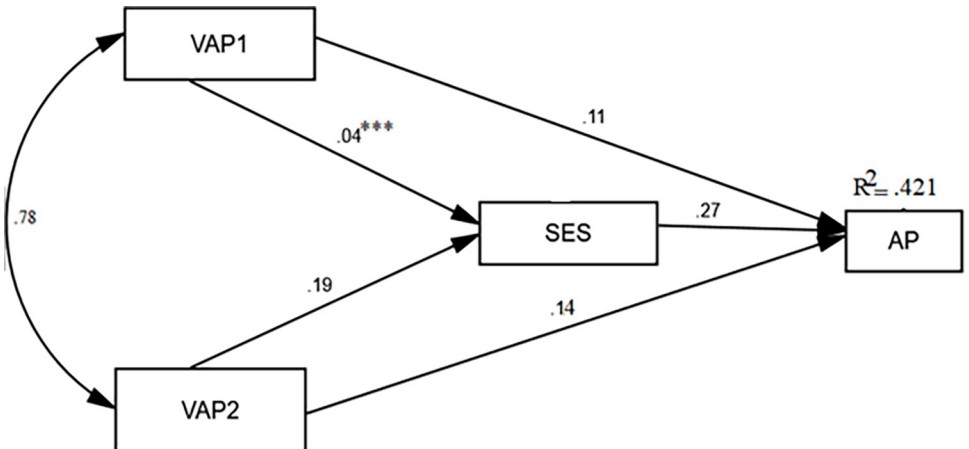

**Fig 1. The association between participating in OOS visual art activities and academic performance with the mediating role of SES.** Note: No asterisk: *P*<0.001, ***not significant, VAP1: participation visual art activities organized and supervised by privately owned training centers, and (VAP2) participation of visual art activities supported and involved by parents in-and-out of home activities.

approximation (RMSEA, 0.11) were highly significant ($P < .001$), which implies that the model fits the data well. According to the SEM or path analysis findings in Fig 1 and Table 4, children's participation in VAP1 and VAP2 determines their academic performance ($\beta = 0.14$, $p < .001$, respectively). Astonishingly, the results determined that SES does not influence the likelihood of children participating in OOSVAP1 ($\beta = 0.04$, $p = .067$). Nevertheless, SES predicts the participation of children in OOSVAP2 ($\beta = 0.19$, $p < .001$). Unsurprisingly, the SEM results indicate that children's SES has a substantial impact on children's academic performance ($\beta = 0.27$, $p < .001$).

## Do economically disadvantaged children benefit from participating in OOS visual art activities?

The bootstrapping method is used to encompass eight models that estimate 95% confidence intervals for these effects from 2000 resamples of the data, and those confidence interval values that do not include zero within the upper and lower bounds imply a significant p-value at less than 0.001. The present SEM finding illuminates that VAP1 has an insignificant indirect effect on academic performance with a mediating role of SES ($\beta = 0.01$, $p = .061$). In contrast, SES

**Table 4. The direct and indirect effect of OOS visual art activities on academic performance.**

| Path | Standardized Effect Size | 95%CI |
|---|---|---|
| VAP1 → AP | .11 | [0.036,0.051] |
| VAP2 → AP | .14 | [0.033,0.062] |
| VAP1 → SES → AP | .12*** | [-0.075, 0.029] |
| VAP2 → SES → AP | .19 | [0.008,0.010] |
| VAP2 → VAP1 → AP | .23 | [0.005,0.015] |
| VAP1 → VAP2 → AP | .22 | [0.035,0.042] |
| VAP1 → VAP2 → SES → AP | .15 | [0.011, 0.031] |

No asterisk = *P*<0.001

***not significant

considerably accounted for the significant indirect impact of VAP1 on academic performance ($\beta$ = 0.05, $p$ <0.001). Moreover, the indirect effect of VAP1 on academic performance through the mediating role of VAP1 and the serial mediating role of VAp1 and SES was significant ($\beta$ = 0.11, $p$ <0.001, $\beta$ = 0.04, $p$ <0.001, respectively).

## Discussion and conclusion

The present study sheds light on Chinese primary school students' likelihood of visual art-based participation outside the school at privately owned schools and activities organized by parents. In addition, this paper addressed whether parents' socioeconomic status determines OOS art participation among students. Ultimately, the study examined its primary objective: to show the association between OOS art participation and academic performance by taking SES. Accordingly, two models (*development and social inequality gap reduction*) were proposed to guide the purpose of the current study.

### Development model

One of the preliminary findings of the present study is that in contemporary China, children and young adolescents' participation in OOS visual art activities is not led by gender. This finding negates a large volume of Western literature that claims that visual art activities are mostly preferred and performed by girls only [7, 19, 51]. One possible explanation is that before the recent one-child policy relaxation, Chinese parents indebted possessing only one child who initiated parenting drawn out from the Confucius ideology, which practiced an autocratic or aggressive parenting style in which children (boy or girl) ought to obey and perform in any life path directed from their parents [35, 40]. On the other hand, children in Western countries have free will to be engaged in the ECAs they enjoy and change or add to other ECAs whenever they want to.

Furthermore, this paper's findings illustrate the moderate impact of parental SES on academic performance. Mothers' educational attainment and monthly income significantly promote children's academic performance in China. A series of studies in China revealed that affluent children outperform their counterparts, given that they hold cultural capital that enables them to access social and academic assets, which promotes academic gaps [11, 40, 48]. Thus, parents must obtain cultural capital to furnish their children's future professional careers, and Chinese mothers are known for mainly taking charge of their children's academic adjustment. Consequently, the study's findings underline that mothers' income is related to the likelihood of attending OOS visual art activities organized and supervised by a commercial training center.

On the other hand, the father's educational attainment and the mother's income determine the extent of children's OOS activities with their parents to broaden their visual art activities. These findings perfectly explain the modern Chinese parenting style, which practices a strategy in which both parents take different but parallel responsibilities for their children's development. Moreover, our analysis supports the development model, which pronounces the influence of ECAs on children's cognitive and noncognitive development [12, 27, 36]. Astonishingly, this finding negates Chinese evidence, which argues that children's participation in after-school ECAs doesn't only predict academic benefits directly but indirectly [11]. However, Tan and colleagues' study has two potential limitations: the study utilized the 2014 China Education Panel Survey (CEPS) and doesn't explain the present dynamic Chinese society, which underwent radical economic and social change. The study assessed ECAs in a dichotomous variable (Yes or No) that doesn't disclose the particular ECA. Similarly, the current research argues with the latest Chinese evidence that only informal ECAs determine

academic performance, but not formal ECAs, which are regularly scheduled and highly structured with a teacher's supervision [7].

In the meantime, the finding of the study is in line with recent evidence in the southeast of the USA that participants, particularly those involved in visual art and music ECAs, enhanced their English language art and mathematics grade points compared to their counterparts from other ECAs such as martial art, dancing, soccer, and so on [5]. Likewise, Lee and Lee's Quasi-experimental study in Kenya affirms that art education was a catalyst for raising the students' autonomy and confidence because it was a discipline that encouraged proactive self-reflection and self-expression [12], which suggests that using visual art as a pedagogy for emotional learning, which aims for students' strong agency and confidence [36]. In light of this paper's exclusive measurement of participation in OOS visual art activities, the findings demonstrate that visual art activities given by privately owned centers and organized by parents in and out of the home have a favorable and substantial impact on children's academic performance [1].

Although there is a large body of evidence that confirms the advantages of undisclosed and unstipulated ECAs on children's well-being [10, 44], the present study adds scanty relevant evidence that highlights the influence of visual art activities on academic achievement [23, 36]. In addition to the overall impact of participation in OOS visual activities on cognitive development, the present study suggests that the effect of visual activities accompanied by parents is more prominent than formal OOS visual art lessons. Likewise, the preceding and latest literature indicate that parents ought to be involved in their children's ECAs so that children foster their cognition and emotion, which strengthens their self-efficacy [9, 12, 24]. Although earlier studies manifest that educated and well-off Chinese parents are immersed in their children's development only through paying their educational and personal needs expenses, the present research unveiled that contemporary Chinese parents embrace a Western parenting style embedded with Chinese culture that nurtures parents to invest their time and earnings on their children OOS ECAs organized by private training centers and activities supervised by parents in-and-out of the home. Such activities that extend the interaction time between children and parents allow parents to support and follow their children's academic progress and time management [1].

## Social inequality gap reduction

It is well documented that OOS ECAs are serviced by commercial training centers that spotlight cultural capital's power in children's well-being [e.g., 1, 25, 27]. Nevertheless, in the modern world, low-SES societies strive to forfeit their will to send their children to OOS centers to enrich their self-confidence among their peers [12, 32]. Afterward, the present study embraces the *Social Inequality Gap Reduction* [10], which is buoyed by an abundance of literature that stresses that economically disadvantaged children benefit from ECAs more than privileged children [32, 34, 38]. In addition, this paper considered OOS activities supervised and organized by parents to enhance children's ECA cognition, emotion, and self-efficacy [12, 15, 27]. Strangely, this paper found that children's SES does not determine the chance of being in structured OOS visual art activities organized by private training schools, which aligns with the latest study assumption [32] and contradicts with Chinese literature that found the positive association between parental SES and the likelihood of their children participating in different ECAs [1, 9, 11]. In fact, a recent study revealed that low-SES students participated in more formal ECAs, while high-SES students were more in informal ECAs than their counterparts [7].

Consequently, the model found that the impact of participating in structured OOS visual art activities at private centers on academic performance cannot be arbitrated or explained by children's SES. This unexpected finding disputes a large body of studies claiming the

significance of OOS ECAs for low-SES children since these children cannot obtain such an environment at home [c.f. 1, 33]. We have two plausible explanations for these unforeseen findings. Mansour and colleagues suggested that visual art activities are ideal for academically high-achieving students who know how to manage their ECAs with their school work [18]. In this research context, the power of SES in China is weightless, given that the one-child policy enables every parent to provide affordable necessities for their child to lay a foundation for their future success. China tracks meritocracy political thought, which indicates that individuals climb the social class ladder based on their achievement or performance [9, 27]. Hence, parents from any socio-class invest their capital in their online well-being and development, which leads them to a resounding endeavor in an academic and future professional career, bringing a high return rate for the parents.

In contrast, although the path analysis finding indicates that SES does not account for the influence of participating in OOS visual art activities organized by private training centers on academic performance, the model reveals that the effect of participating in OOS visual art activities organized by parents on children's academic performance is noteworthy for advantaged children from the upper class. Interestingly, the finding contradicts previous claims about the non-involvement of affluent Chinese parents in spending time with their children through in-and-out-of-home ECAs to enhance their children's development and parent-child attachment [1, 2, 27]. This finding illuminates that low-SES parents lack time or energy, as their jobs have long working hours that result in physical fatigue. Educated parents have fixed working hours that permit them to use their off and weekend time in activities with their children. In the preceding literature, Chinese parents are known for spending money rather than time on their children's cognitive and noncognitive adjustment. This paper confirms the latest studies' perspective, arguing that present-day maternities follow their children's academic and nonacademic progress by involving themselves and their monetary allegiance [e.g., 12, 27, 40]. The aforementioned study mentioned that in addition to participation in OOS-organized visual art programs, parents should support their children by expanding their exposure to activities ([.f., 1].

Moreover, a longitudinal study revealed that children who spend OOS art activities with their parents showed marvelous performance in math and reading [26]. Likewise, later evidence confirmed that visiting art exhibitions, museums, and galleries boosts children's motivation and efficacy [33, 36, 44]. This explains why the relationship between visual art participation and academic performance is nourished by parental involvement [1, 2, 24].

## Conclusion

The present study sought to examine the impact of OOS visual art activities on academic performance by considering the SES through the theoretical lens of the development and social inequality gap reduction models. The study's findings suggested that the participation of children and adolescents in visual art activities in private training centers and parent supervision in and out of the home develops educational outcomes. Most interestingly, this investigation claimed that SES doesn't determine the likelihood of children being involved in visual art activities in private art education centers or parental supervision and organized in and out of the home. Hence, the study noted that the social inequality gap reduction model may not apply to contemporary Chinese society as the dynamic SES distribution.

### Implications, limitations, and direction for future studies

Based on our first empirical study conducted to examine the effect of participation in OOS visual art activities organized by privately-owned centers and organized by parents in and out

of home on children's cognitive development in view of children's SES, we concocted promising and robust theoretical policy and practical implications. (1) By considering the patent impact of visual art activities in building a generation fueled with solid creativity, self-efficacy, self-concept, and communication skills, policy-makers in developing countries that are under radical economic development, such as China, ought to cherish the application of visual art activities by introducing it under a formal curriculum [5], so that qualified and trained teachers can integrate art into their subject matter. (2) This study suggests that government policy-makers and educators have to strive hard to monitor the cost of OOS private training centers to eliminate social injustice from the educational system; or like Western countries, government, non-government organizations, and communities in developing countries should work together to promote free or low-cost OOS training centers that provide different ECAs for children and adolescents. (3) The present study sheds light on shifting the mindset of parents who perceive spending money on their children as their best effort with zero parent-child communication and relationship time. (4) Most importantly, this study proposes that the *social inequality gap reduction model* is impractical for modern-day Chinese society, given that the one-child policy's long-term effect made SES impotent in accessing cultural capital [11].

The current study findings and implications should be grasped in light of the limitations that could not be addressed. The study's first limitation is that the sample was obtained from primary school students in one province, so the generalization radius of our findings does not include students of early childhood and late adolescent age. Second, since the study is limited to applying only one ECA, the results do not represent other ECAs in children's cognitive development. Third, the study's findings cannot be generalized to Western countries or other countries, so we kindly stimulate scholars and educators from developing countries to underline the power of visual art activities in children's psychological and cognitive well-being.

## Supporting information

**S1 File.**
(SAV)

## Acknowledgments

We want to express our great appreciation and admiration to the adorable children who were part of this study and their parents and teachers who made the data collection happen.

## Author Contributions

**Conceptualization:** Genman Deer, Endale Tadesse, Sabika Khalid, Chunhai Gao.

**Data curation:** Hao Wu, Li Zhang, Endale Tadesse, Congyu Duan, Chunhai Gao.

**Formal analysis:** Genman Deer, Hao Wu, Li Zhang, Endale Tadesse, Congyu Duan, Wang Tian.

**Investigation:** Genman Deer, Hao Wu, Li Zhang, Sabika Khalid, Wang Tian, Chunhai Gao.

**Methodology:** Genman Deer, Hao Wu, Li Zhang, Sabika Khalid, Chunhai Gao.

**Resources:** Chunhai Gao.

**Supervision:** Hao Wu, Chunhai Gao.

**Validation:** Hao Wu, Li Zhang, Endale Tadesse.

**Visualization:** Endale Tadesse, Congyu Duan.

**Writing – original draft:** Genman Deer, Endale Tadesse, Sabika Khalid, Wang Tian, Chunhai Gao.

**Writing – review & editing:** Hao Wu, Li Zhang, Congyu Duan.

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
