## [Editor Report · Decision Letter 0]

19 Jul 2023

PONE-D-23-15066Does Parental time or money spent on children's visual art activities close the academic performance gap? Evidence from contemporary ChinaPLOS ONE

Dear Dr. Gao,

Thank you for submitting your manuscript to PLOS ONE. After careful consideration, we feel that it has merit but does not fully meet PLOS ONE’s publication criteria as it currently stands. Therefore, we invite you to submit a revised version of the manuscript that addresses the points raised during the review process.

We look forward to receiving your revised manuscript.

Kind regards,

Najmul Hasan, PhD

Academic Editor

PLOS ONE

4. Please ensure that you include a title page within your main document. You should list all authors and all affiliations as per our author instructions and clearly indicate the corresponding author.

Additional Editor Comments:

Manuscript ID: PONE-D-23-15066

Does Parental time or money spent on children's visual art activities close the academic performance gap? Evidence from Contemporary China

This paper is about parental time or money spent on children's visual art activities applying a cross-sectional survey study and then analyzing it with a quantitative approach. The topic is relevant and very up-to-date in the Social and Behavioral Sciences field. In addition, the study is also applicable to the mission of the journal in highlighting children’s behavioral studies. However, in my opinion, there are some flaws with this paper, which I address in turn below. I show below some ideas for the author on how to make the paper even more robust.

1. The title needs to be clearer. The word “contemporary” looks odd.

2. The submission has good identification of the problems/issues related to the topic, the topic's importance, and the research gap. Though it is a rigorous study in the context of cross-sectional data, the study has lacking research aim/purpose as well as the research question. What actually author(s) want to answer the question by conducting this research and how this study contributes to the literature and practically – children/parents? What’s new in this study? compared to others?

3. The literature review sounds good. But I am worried about many old unnecessary citations. I strongly suggest to the author(s) reduce the unnecessary citation.

4. Though the methodology looks good, and the author(s) applied random sampling for this study, is there any specific framework for determining sample size? Also, I appreciate the ethical approval number with evidence. I also expect the details of the data collection procedure.

5. The results are supported by the correct data analysis, they are sufficient.

6. Some minor revisions are needed, specifically in the section on implications. The implications are not conclusions significantly. Though the research findings support the results to be classified into practical implications, the theoretical contribution is lackings.

7. The manuscript still not yet representing good academic writing and needs language editing.

In summary, please note that I am only trying to point out that your paper does not satisfy the journal's quality requirements at this stage of development, and further MAJOR Revision is needed —however, exciting topic. I hope my comments will help the authors improve the article. Good luck.

<quillbot-extension-portal></quillbot-extension-portal><quillbot-extension-portal></quillbot-extension-portal>

---

## [Author Response · Author response to Decision Letter 0]

8 Sep 2023

Point-by-Point Response to Reviewers’ Comments

Dear Editors and reviewers, 

I am writing this letter on behalf of my coauthor regarding our Manuscript ID: PONE-D-23-15066, entitled " Effect of out-of-school Visual Art Activities on Academic Performance. The Mediating Role of Socioeconomic Status." We want to thank the respected editors and reviewers for providing constructive comments and suggestions for the manuscript to be considered for publication with minor adjustments. As you can see below, we have responded to all reviewer's inquiries before the manuscript is published.

Comments

Comment: Does Parental time or money spent on children's visual art activities close the academic performance gap? Evidence from Contemporary China

Author Response: We agree with the respected reviewer’s comment and refined the revised manuscript title.

Comment: The submission has good identification of the problems/issues related to the topic, the topic's importance, and the research gap. Though it is a rigorous study in the context of cross-sectional data, the study has lacking research aim/purpose as well as the research question. What actually author(s) want to answer the question by conducting this research and how this study contributes to the literature and practically – children/parents? What’s new in this study? compared to others?

Author Response: Please refer to pages 4, 5, and 11, which demonstrate the study's contribution and the research hypothesis.

Comment: The literature review sounds good. But I am worried about many old unnecessary citations. I strongly suggest to the author(s) reduce the unnecessary citation.Authors Response: We kindly want to inform the reviewer that, putting full names would take make the organization of the manuscript out of format, messy and hard to follow.

Author Response: It was a great suggestion. Necessary amendments are addressed.

Comment: Though the methodology looks good, and the author(s) applied random sampling for this study, is there any specific framework for determining sample size? Also, I appreciate the ethical approval number with evidence. I also expect the details of the data collection procedure.

Author Response: The suggestion is addressed.

Comment: Though the research findings support the results to be classified into practical implications, the theoretical contribution is lackings.

Author Response: Based on the findings of the study. We addressed the theoretical implication of the study, which can be exhibited in the revised manuscript.

Comment: The manuscript still not yet representing good academic writing and needs language editing.

Author Response: A professional Academic editing service has been used to ensure the language accuracy of the manuscript.

---

## [Decision Letter · Decision Letter 1]

12 Dec 2023

PONE-D-23-15066R1Effect of out-of-school Visual Art Activities on Academic Performance. The Mediating Role of Socioeconomic StatusPLOS ONE

Dear Dr. Wu,

Thank you for submitting your manuscript to PLOS ONE. After careful consideration, we feel that it has merit but does not fully meet PLOS ONE’s publication criteria as it currently stands. Therefore, we invite you to submit a revised version of the manuscript that addresses the points raised during the review process.

We look forward to receiving your revised manuscript.

Kind regards,

Najmul Hasan, PhD

Academic Editor

PLOS ONE

Reviewers' comments:

Reviewer's Responses to Questions

**Comments to the Author**

1. If the authors have adequately addressed your comments raised in a previous round of review and you feel that this manuscript is now acceptable for publication, you may indicate that here to bypass the “Comments to the Author” section, enter your conflict of interest statement in the “Confidential to Editor” section, and submit your "Accept" recommendation.

Reviewer #1: (No Response)

Reviewer #2: (No Response)

2. Is the manuscript technically sound, and do the data support the conclusions?

Reviewer #1: Partly

Reviewer #2: Yes

3. Has the statistical analysis been performed appropriately and rigorously? 

Reviewer #1: No

Reviewer #2: Yes

4. Have the authors made all data underlying the findings in their manuscript fully available?

Reviewer #1: No

Reviewer #2: Yes

5. Is the manuscript presented in an intelligible fashion and written in standard English?

Reviewer #1: No

Reviewer #2: Yes

6. Review Comments to the Author

Reviewer #1: In this paper, authors explored the impact of money and time spending of parents on children's academic activities. There is ample space for improvement with following suggestions

Abstract

Abstract should be equipped with used methodology and policy recommendations.

Introduction

This section fails to explain the concepts of study, novelty and significance of the study. Excessive citation of earlier literature is not appreciated because it omits the contribution of the study. Authors just quoted the studies in the introduction instead of case building. This section should be enriched with empirical and theoretical evidences on the topic. Moreover, authors should consult latest studies on the topic to convince the readers about suitability of the topic to be explored further.

Literature Review

This section should amplify the deficiency in the earlier literature that is going to met by this study.

Each hypothesis should be established with the help of earlier theories and empirical evidences.

Method

The used methodology MANOWA should be explained in this section.

I would like to suggest incorporate the tables and figure in the main body of the literature where these are suitable.

Paper should be restructured as per standard academic pattern.

Conclusion section is not present in the study.

Reviewer #2: I don't think the authors addressed the previous round of reviewer comments fully. This is evidenced by

1. No research questions/objectives are added

2. Sample size determination was not added

3. There were still many redundant citations (sometimes three) in many sentences.

I would like to have more information about the research procedure. The data come from three sources, the parents (education and income of the parents), the child (participation in art activities), and the schools (students' academic scores). How is the data of the parents obtained and how is it matched with the corresponding child? For this kind of sensitive information, verbal consent was insufficient. Did the parents know that the child's academic scores will be obtained from the schools and provided to the researchers?

I would like to see a written information sheet that was provided to the parents to see if sufficient information about the study and the collection of students' academic scores was given.

7. PLOS authors have the option to publish the peer review history of their article (what does this mean?). If published, this will include your full peer review and any attached files.

Reviewer #1: No

Reviewer #2: No

---

## [Author Response · Author response to Decision Letter 1]

18 Jan 2024

Cover Letter

September 13, 2023

Dear Editors and reviewers, 

I am writing this letter on behalf of my coauthor regarding our manuscript PONE-D-23-15066R1, entitled " Effect of out-of-school Visual Art Activities on Academic Performance. The Mediating Role of Socioeconomic Status". We thank the respected editors and reviewers for providing constructive comments and suggestions for the manuscript to be considered for publication with minor adjustments. As you can see below, we strived to attempt every inquiry raised by the three reviewers before the manuscript was ready for publication.

Reviewer 1 Comments

Comment 1: Abstract should be equipped with used methodology and policy recommendations.

Author Response: Thank you for the compliment on the rigorous review the authors did. In addition, per the reviewer's suggestion, we included relevant and evidence-based implications and the methodology employed in the present study.

Comment 2: This section fails to explain the concepts of study, novelty and significance of the study.

Author Response: The revised introduction section of the manuscript ensured that relevant and recent evidence has been used to strengthen the argument and highlight the particular purpose of the study.

Comment 3: The used methodology MANOWA should be explained in this section

Author Response: Relevant explanations in the method section provide vivid information for the reader.

Comment 4: Conclusion section is not present in the study.

Author Response: Thank you for the significant insight, which avoids confusion. The Conclusion section has been addressed to provide a comprehensive reflection of the study.

Reviewer 2 Comments

Comment 1: No research questions/objectives are added

Authors Response: Thank you for the suggestion. We have made sure that the main purpose of the study is highlighted at the end of the introduction and literature review section, and regarding the research questions, under the present study section, all the research hypotheses or questions are stated. Kindly refer to the revised manuscript.

Comment 2: Sample size determination was not added

Author's Response: Thank you for the concern. We provided relevant explanations for that. 

Comment 3: How is the data of the parents obtained and how is it matched with the corresponding child? For this kind of sensitive information, verbal consent was insufficient. Did the parents know that the child's academic scores will be obtained from the schools and provided to the researchers?

Author Response: Verbal consent was a supporting means we used to explain the objective of our study for those parents who wanted to give verbal consent; however, written consent was obtained from the rest. In addition, as we mentioned in the methodology section, the teachers and school principals helped us obtain each sampled student's academic performance, and the teachers requested participants to obtain their parental educational and monthly income with consent.

---

## [Decision Letter · Decision Letter 2]

1 Feb 2024

Effect of out-of-school Visual Art Activities on Academic Performance. The Mediating Role of Socioeconomic Status

PONE-D-23-15066R2

Dear Dr. Wu

We’re pleased to inform you that your manuscript has been judged scientifically suitable for publication and will be formally accepted for publication once it meets all outstanding technical requirements.

Kind regards,

Najmul Hasan, PhD

Academic Editor

PLOS ONE

Additional Editor Comments (optional):

Reviewers' comments:

Reviewer's Responses to Questions

**Comments to the Author**

1. If the authors have adequately addressed your comments raised in a previous round of review and you feel that this manuscript is now acceptable for publication, you may indicate that here to bypass the “Comments to the Author” section, enter your conflict of interest statement in the “Confidential to Editor” section, and submit your "Accept" recommendation.

Reviewer #1: All comments have been addressed

Reviewer #2: All comments have been addressed

2. Is the manuscript technically sound, and do the data support the conclusions?

Reviewer #1: Yes

Reviewer #2: Yes

3. Has the statistical analysis been performed appropriately and rigorously? 

Reviewer #1: Yes

Reviewer #2: Yes

4. Have the authors made all data underlying the findings in their manuscript fully available?

Reviewer #1: Yes

Reviewer #2: Yes

5. Is the manuscript presented in an intelligible fashion and written in standard English?

Reviewer #1: Yes

Reviewer #2: Yes

6. Review Comments to the Author

Reviewer #1: The authors have addressed all the comments in the manuscript. Authors put their maximum to improv the quality of the manuscript.

Reviewer #2: (No Response)

7. PLOS authors have the option to publish the peer review history of their article (what does this mean?). If published, this will include your full peer review and any attached files.

Reviewer #1: No

Reviewer #2: No

---

## [Editor Report · Acceptance letter]

26 Apr 2024

PONE-D-23-15066R2 

PLOS ONE

Dear Dr. Wu, 

I'm pleased to inform you that your manuscript has been deemed suitable for publication in PLOS ONE. Congratulations! Your manuscript is now being handed over to our production team.

Kind regards, 

on behalf of

Dr. Najmul Hasan 

Academic Editor

PLOS ONE